# Effect of Tocilizumab on Mortality in Patients with SARS-CoV-2 Pneumonia Caused by Delta or Omicron Variants: A Propensity-Matched Analysis in Nimes University Hospital, France

**DOI:** 10.3390/antibiotics12010088

**Published:** 2023-01-04

**Authors:** Paul Laffont-Lozes, Didier Laureillard, Paul Loubet, Robin Stephan, Myriam Chiaruzzi, Edouard Clemmer, Aurelie Martin, Claire Roger, Laurent Muller, Pierre-Géraud Claret, Radjiv Goulabchand, Clarisse Roux, Jean-Philippe Lavigne, Albert Sotto, Romaric Larcher

**Affiliations:** 1Department of Pharmacy, Nimes University Hospital, Place du Professeur Robert Debre, 30000 Nimes, France; 2Infectious and Tropical Diseases Department, Nimes University Hospital, Place du Professeur Robert Debre, 30000 Nimes, France; 3Department of Microbiology and Hospital Hygiene, Nimes University Hospital, Place du Professeur Robert Debre, 30000 Nimes, France; 4Anesthesiology and Critical Care Medicine Department, Nimes University Hospital, Place du Professeur Robert Debre, 30000 Nimes, France; 5Emergency Medicine Department, Nimes University Hospital, Place du Professeur Robert Debre, 30000 Nimes, France; 6Department of Internal Medicine, Nimes University Hospital, Place du Professeur Robert Debre, 30000 Nimes, France; 7VBIC (Bacterial Virulence and Chronic Infection), INSERM (French Institute of Health and Medical Research), Montpellier University, 34090 Montpellier, France; 8PhyMedExp (Physiology and Experimental Medicine), INSERM (French Institute of Health and Medical Research), CNRS (French National Centre for Scientific Research), University of Montpellier, 34090 Montpellier, France

**Keywords:** COVID-19, variant of concern, interleukin-6 receptor antagonist, early administration, mortality rates

## Abstract

We aimed to assess the factors associated with mortality in patients treated with tocilizumab for a SARS-CoV-2 pneumonia due to the delta or omicron variants of concern (VOC) and detect an effect of tocilizumab on mortality. We conducted a prospective cohort study in a tertiary hospital from 1 August 2021 to 31 March 2022 including patients with severe COVID-19, treated with tocilizumab. Factors associated with mortality were assessed in a Cox model; then, the 60-day mortality rates of COVID-19 patients treated with standard of care (SoC) +/− tocilizumab were compared after 1:1 propensity score matching. The mortality rate was 22% (N = 26/118) and was similar between delta and omicron cases (*p* = 0.6). The factors independently associated with mortality were age (HR 1.06; 95% CI (1.02–1.11), *p* = 0.002), Charlson index (HR 1.33; 95% CI (1.11–1.6), *p* = 0.002), WHO-CPS (HR 2.56; 95% CI (1.07–6.22) *p* = 0.03), and tocilizumab infusion within the first 48 h following hospital admission (HR 0.37, 95% CI (0.14–0.97), *p* = 0.04). No significant differences in mortality between the tocilizumab plus SoC and SoC alone groups (*p* = 0.5) were highlighted. However, the patients treated with tocilizumab within the 48 h following hospital admission had better survival (*p* = 0.04). In conclusion, our results suggested a protective effect on mortality of the early administration of tocilizumab in patients with severe COVID-19 regardless of the VOC involved.

## 1. Introduction

Soon after the beginning of the pandemic, the pathogenicity of the severe acute respiratory syndrome coronavirus 2 (SARS-CoV-2) was reported to be related to a “Cytokine storm” [1]. This deregulation in cytokine secretion was mainly characterized by an extensive expression of interleukin-6 (IL-6) and tumor necrosis factor α (TNF-α) which injure the lungs and other organs to a lesser extent [1,2]. Unsurprisingly, corticosteroids were the first immunomodulatory medications to demonstrate a mortality benefit among patients with coronavirus disease 2019 (COVID-19) and are now a cornerstone of COVID-19 treatment in patients receiving respiratory support [3].

Early on, the potential benefit of IL-6 receptor antagonist monoclonal antibodies such as tocilizumab was suggested [2]. Indeed, since IL-6 was identified as one of the key cytokines involved in the COVID-19-induced cytokine storm, tocilizumab has been proposed to block the downstream signal transduction by binding IL-6 receptors. Thus, it blocks the JAK/STAT tyrosine kinase system and the Ras/mitogen activated protein kinase (MAPK)/NF-κB-IL-6 pathway, thereby limiting the cytokine storm and its life-threatening consequences [4].

In mid-2021, two large platform trials reported that tocilizumab reduced mortality and the use of invasive mechanical ventilation, and increased the chances of a successful hospital discharge [5,6]. Subsequently, numerous studies have investigated its effect in patients with SARS-CoV-2 pneumonia caused by the alpha, beta or gamma variants of concern (VOC) [7,8,9,10,11], even recently reporting that the antagonist of IL-6 receptors also improved long-term outcomes [12]. Unfortunately, in numerous countries, several factors have limited the use of tocilizumab, such as drug shortages, high cost or late approval (December 2021 in Europe, December 2022 in the USA). Therefore, real-life data on tocilizumab treatment remain scarce, especially in patients infected with delta or omicron VOCs [13,14,15]. Moreover, its use is questionable in omicron cases. Indeed, the reduced intrinsic virulence of the omicron VOC and immunization might contribute to decrease the inflammatory response, reducing the COVID-19 severity, and accordingly reducing the benefit of tocilizumab treatment [16].

This study aimed to report real-life data on tocilizumab use in patients with SARS-CoV-2 pneumonia caused by the delta or omicron VOC. The main objective was to assess factors associated with mortality in patients treated with tocilizumab plus standard of care (SoC). Then, we aimed to compare the prognosis of patients treated with tocilizumab plus SoC to those treated with SoC alone, to detect a potential effect of tocilizumab on mortality.

## 2. Results

### 2.1. Population

Characteristics of the study population are reported in Table 1.

Among the 1213 patients admitted for COVID-19 during the study period, 118 patients (9.7%) were treated with tocilizumab and included in the study (Figure 1). Of them, 67% were males (*n* = 79), and the median age and Charlson index were 69 years old (IQR, 56–77) and 1 (IQR, 0–2), respectively.

At hospital admission, the World Health Organization Clinical Progression Scale (WHO-CPS) was 5 in 68 patients (58%) and 6 in 50 patients (42%). The median C-reactive protein (CRP) level was 124 mg/L (IQR, 89–165), and the pulmonary involvement on CT-Scan was classified as moderate or severe in more than 80% of the patients. A total of 101 patients (86%) were infected with delta, and 17 (14%) with omicron.

Tocilizumab was administered at a median dose of 600 mg (IQR, 600–800), and 68 patients (58%) received their first tocilizumab dose within the first 48 h of hospital stay. Only six patients (5%) had a second dose.

### 2.2. Outcomes

The median CRP decreased to 47 mg/L (IQR, 23–89) following the tocilizumab infusion. The decrease in CRP was shown in all patients but 9, and the WHO-CPS increased in 34 patients (29%) (Figure 2).

The median length of hospital stay was 10 days (IQR, 7–15). At the end of the follow-up period of 60 days, 88 patients (75%) were discharge alive (WHO-CPS ≤ 3), 4 patients (3%) were still hospitalized with a WHO-CPS score of 4, 5, 6 and 9, respectively, and 26 patients (22%) had died (Table 1).

Importantly, the 60-day survival probability was at 87%, 95% CI (79–95%) in patients treated with tocilizumab within the first 48 h following hospital admission, and at 66%, 95% CI (54–80%) in those treated later (*p* = 0.009). Moreover, the 60-day survival probability was not statistically different in delta and omicron cases (77%, 95% CI (69–86%) versus 82%, 95% CI (66–100%), *p* = 0.6), as illustrated in Figure 3.

### 2.3. Factors Associated with Mortality

Results of the univariate and multivariable Cox regressions are presented in Table 2.

In the multivariable analysis, age (HR 1.06; 95% CI (1.02–1.11), *p* = 0.002), Charlson index (HR 1.33; 95% CI (1.11–1.60), *p* = 0.002) and WHO-CPS score at the time of tocilizumab infusion (HR 2.58; 95% CI (1.15–1.66), *p* = 0.03) were independently associated with mortality. In contrast, early administration of tocilizumab had a protective effect (HR 0.37; 95% CI (0.14–0.97), *p* = 0.04).

### 2.4. Tocilizumab Treatment Effect on Mortality

Using propensity score matching in a Cox model, tocilizumab treatment was shown to have no effect on mortality (*p* = 0.5)**.** However, as reported in Figure 3, the 60-day survival probability was higher in patients treated with tocilizumab within the first 48 h of their hospital stay compared to those treated later and those treated with SoC alone (87%, 95% CI (79–95%) vs. 66%, 95% CI (54–80%) vs. 75%, 95% CI (68–84%), *p* = 0.04).

In addition, when we included variables resulting in an imbalance between the groups after propensity score matching (namely, age, high-flow oxygen support needs and mechanical ventilation needs), the effect of tocilizumab was significant in the adjusted Cox model (HR 0.48, 95% CI (0.27–0.86), *p* = 0.01).

Of note, COVID-19 associated pulmonary aspergillosis (CAPA) was diagnosed in 3 patients among those patients treated with tocilizumab (3/118; 3%) and in 6 among those treated with SoC alone (6/706; 1%). All of them required mechanical ventilation.

## 3. Discussion

This study reported the results of a single-center cohort that included 118 patients with severe COVID-19 treated with tocilizumab on top of SoC and described their clinical features and outcomes in a period of the circulation of the delta and omicron variants. We found a 22% overall 60-day mortality rate, similar in both omicron and delta cases. A higher age, higher Charlson index and higher level of oxygen support were independently associated with mortality. In contrast, the early administration of tocilizumab was independently associated with better survival. Importantly, after propensity score matching, this study did not find a significant effect for tocilizumab + SoC on 60-day mortality compared to SoC alone. However, the effect of tocilizumab was significant after adjustment for variables resulting in an imbalance between the groups.

In our cohort of patients treated with tocilizumab, as has been widely reported in numerous previous studies on COVID-19 patients [17], mortality was independently associated with higher age and a higher Charlson index. Among comorbidities, diabetes has been reported as a critical risk factor for severe SARS-CoV-2 infection and death [18], as highlighted by our result (death rates of 43% vs. 23%). Similarly, the level of hypoxemia assessed by PaO_2_/FiO_2_, the SpO_2_/FiO_2_ ratio or the WHO-CPS is one of the most significant factors previously reported to drive mortality in COVID-19 patients [10].

Other studies [7,8,10,11,19], including platform trials [5,6], have reported a favorable effect of tocilizumab in alpha, beta or gamma cases; however, the benefit of tocilizumab in patients infected with delta, which is reported to be more virulent [20], or in patients infected with omicron, which is, in contrast, reputed to be less virulent [21], remained disputable. To the best of our knowledge, this study was the first to report the use of tocilizumab in a large cohort of severe COVID-19 patients infected with the delta and omicron VOC, suggesting that, among the patients developing a severe form of the disease, the mortality rates were similar in the omicron and delta cases, as it has been reported with previous VOCs [14]. In the same line, some authors have highlighted that the rate of severe COVID-19 may change depending on the VOC, but, in cases of severe forms, they reported that the cytokine secretion remained stable and independent from the VOC [22]. Accordingly, our results suggested that tocilizumab remained an efficient option for the treatment of severe COVID-19 patients in association with corticosteroids, regardless of the VOC involved.

One striking finding of this study was that early administration of tocilizumab (within 48 h following hospital admission) was associated with a better outcome. This result was in accordance with those reported in other studies suggesting the benefit of tocilizumab administration within 2–3 days of hospital admission [8,10,11]. Similarly, the results of the sensitivity analyses of RECOVERY trials [5] have also suggested that patients who received tocilizumab within seven days of symptom onset are those who benefited the most. It is worth noting that some authors [23] have highlighted that tocilizumab’s effect on mortality is more related to its rapid administration after the onset of oxygen requirement than after symptom onset. Accordingly, the probability of any benefit of tocilizumab has been estimated at 99%, 96% and 75%, for patient receiving simple oxygen, non-invasive ventilation and mechanical ventilation, respectively [23]. Our results, consistent with these studies, suggest that administering tocilizumab as soon as a COVID-19 patient requires oxygen support may provide the greatest survival benefit.

However, after two years of the pandemic, tocilizumab remains a debated therapeutic option for some physicians caring for COVID-19 patients. First, conflicting results [24,25,26] discouraged some of them from using it. Second, tocilizumab has been reported to be associated with an increased risk of infections [27], and especially an increased risk of CAPA [28]. However, COVID-19 itself and corticosteroids also increased this risk [29,30], whereas few superinfections, especially CAPA, have been reported in large trials [5,6]. In accordance with these reports, during the study period, only 12 COVID-19 patients were diagnosed with proven, probable or possible CAPA [31] in our center, and among them only 3 had been treated with tocilizumab. Moreover, despite this possible increased risk of health-care associated infections, the benefits of tocilizumab use seem to outweigh the risks in severe COVID-19 patients, since it decreased all-cause mortality in the short [5,6] and long term [12].

Some data have suggested a second dose of tocilizumab in the case of weak clinical improvement [5]. However, the patients who could benefit from a second dose remain poorly identified. In our cohort, 6 patients who had increased their oxygen requirements 12–24 h after a first dose of tocilizumab subsequently received a second dose. All of them were discharged alive. Moreover, in our study, among the 9 patients whose CRP increased after a first infusion of tocilizumab, none received a second dose, and 4 (44%) of them died. Along the same lines, Khurshid et al. reported that the COVID-19 patients who maintained higher CRP values during treatment had the worst outcomes [32]. However, further exploration is mandatory to assess the use of biomarkers, such as CRP, to trigger a second dose of tocilizumab.

Our study has both strengths and limitations. First, our conclusions are limited by the study’s monocentric design, which could induce bias in the interpretation of the results and limit their generalization. However, this bias is limited by the standardization of care for COVID-19 patients in accordance to the WHO international guidelines [33]. Moreover, all patients received dexamethasone, which may synergistically interact with tocilizumab [34]. Second, the data were retrospectively collected for our control group, which could induce bias in data collection. Third, we performed a propensity score analysis and found that tocilizumab had no effect on mortality. However, the relatively small size of the cohort could have limited the detection of its effect on mortality. Indeed, based on previously published data [5,6], we should have included 3902 patients in the study (alpha = 0.05 and power 90%) to detect a 4% reduction in mortality. Finally, we reported mortality rates similar to those of randomized clinical trials [5,6,12]; however, in our cohort, the patients were older [35] and had more severe cases (according to our institutional guidelines, tocilizumab was given to those with rapid increase in oxygen requirements), which could have increased mortality rates.

## 4. Materials and Methods

### 4.1. Study Design and Settings

This prospective, monocentric, observational cohort study was carried out in a French University Hospital from 1 August 2021 to 31 March 2022.

In our hospital, during the study period, the bed capacity for COVID-19 patients care ranged from 9 to 81 ward-beds and from 41 to 81 ICU-beds.

### 4.2. Tocilizumab Treatment

Tocilizumab was first approved by the European Medicines Agency (EMA) in 2009 for the treatment of rheumatoid arthritis and juvenile idiopathic arthritis. In June 2021, tocilizumab was granted an emergency use authorization (EUA) for the treatment of COVID-19 in the United States, and the EMA recommended tocilizumab for adults with COVID-19 who are receiving systemic treatment with corticosteroids and require supplemental oxygen or mechanical ventilation in December 2021. More recently, in December 2022, the Food and Drug Administration (FDA) approved tocilizumab for patients with severe COVID-19.

Taking into account the body of evidence available in the literature [5,6], the Anti-Infective Drugs Committee of the Nimes University Hospital authorized the off-label use of tocilizumab in the treatment of severe COVID-19 in June 2021. Thus, tocilizumab started to be administered at physicians’ discretion to patients treated with SoC for severe or critical COVID-19 (i.e., intravenous dexamethasone 6mg once a day, prophylactic heparin and oxygen therapy to maintain SpO_2_ > 94%). Importantly, remdesivir was not recommended in our center, according to international [33] and national guidelines [36].

The detailed criteria for tocilizumab initiation were as follows: (1) CRP ≥ 75 mg/L and an increase of oxygen needs ≥ 2L/min within 48 h, or (2) lack of improvement after 48 h of SoC or (3) ICU admission ≤ 24 h.

Tocilizumab was administered intravenously at a dose depending on bodyweight: 800 mg if weight > 90 kg; 600 mg if weight > 65 kg and ≤90 kg; 400 mg if weight > 40 kg and ≤65 kg; and 8 mg/kg if weight ≤ 40 kg [5]. A second dose could be given 12–24 h later at the clinician’s discretion if the patient’s condition had not improved.

Contraindications for tocilizumab were untreated confirmed or suspected bacterial or fungal infection, hepatic cytolysis > 5 upper limit of normal (ULN), neutropenia < 0.5 cells × 10^9^/L and thrombopenia < 50 cells × 10^9^/L [5].

### 4.3. Patients

All patients hospitalized during the study period and receiving tocilizumab were screened daily for inclusion using the software of the Pharmacy Department (Pharma^®^, Computer Engineering, Paris, France). Then, those patients receiving tocilizumab for severe or critical COVID-19 were included in the study. Patients younger than 18 years old or pregnant, those treated with tocilizumab for a reason other than COVID-19 and those with a hospital stay of a length ≤ 1 day were excluded (Figure 1). Severe COVID-19 was defined as previously described [37,38].

All consecutive adult COVID-19 patients hospitalized during the study period for severe SARS-CoV-2 pneumonia and treated with SoC were retrieved by screening the hospital database using the International Classification of Diseases [39] to build a control group. Patients younger than 18 years old or pregnant, those treated with tocilizumab and those with a hospital stay of a length ≤ 1 day were excluded (Figure 1).

### 4.4. Data Collection

In patients treated with tocilizumab, their demographical data and morbidities were collected prospectively at hospital admission, and their Charlson index was calculated [40]. The percentage of pulmonary involvement was assessed on a chest CT-Scan at admission and was classified as previously described [41] by a radiologist and confirmed blindly by an infectious disease physician trained in chest CT-Scan interpretation (R.L.). SARS-CoV-2 VOC were also recorded after detection by genotyping or gene sequencing on nasopharyngeal swabs. CRP was recorded 24–48 h before and after tocilizumab infusion. The severity of respiratory failure was assessed using the WHO-CPS at admission and at the time of tocilizumab infusion [42]. In the control group, the demographical data, morbidities, oxygen requirements, VOC and outcomes were collected retrospectively.

### 4.5. Outcomes

The lengths of the ICU and hospital stays were recorded. The respiratory status was assessed using the WHO-CPS at 28 and 60 days after hospital admission as recommended by the WHO working group on the clinical characterization and management of COVID-19 infection [42]. Vital statuses at ICU and hospital discharge and 60 days after hospital admission were collected; then, mortality rates were calculated. Additionally, cases of CAPA have been retrospectively collected [28].

### 4.6. Statistical Analysis

Categorical data were described as numbers and percentages, and continuous data as medians with 25th and 75th percentiles (interquartile range: IQR). The population was divided into two groups according to vital status at 60 days. The categorical variables were compared by Chi-square or Fisher’s exact test, and the continuous variables were compared by Student’s t test or Wilcoxon’s rank-sum test as appropriate.

Factors associated with 60-day mortality were assessed using univariable and multivariable cox regression model. Factors associated with 60-day mortality in the univariate analysis (cut-off of *p* ≤ 0.2) were included in the multivariable analysis. Proportional hazard assumption was assessed by inspecting the scaled Schoenfeld residuals. Results of Cox regression model were reported as hazard ratio (HR) with 95% confidence interval (95% CI).

Propensity score matching was also performed to compare COVID-19 patients treated with tocilizumab and SoC with those treated with SoC alone. Patients were matched (1:1) with the algorithm for nearest-neighbor matching without replacement, using a maximum tolerance distance between the matched subjects of 0.1 standard deviation. The confounding variables used to calculate the propensity scores were age, BMI ≥ 30 kg/m^2^, ICU admission needs, non-invasive high-flow extra-oxygen support needs, mechanical ventilation needs, previous immunosuppressive treatment (including corticosteroids) and previous chemotherapy, as well as each variable included in the Charlson index, its value and the VOC. We identified the variables resulting in an imbalance between the groups after propensity score matching by calculating the standardized mean difference; then, we included these in the subsequent Cox proportional hazards models as covariates to assess the effect of tocilizumab treatment.

Survival curves were obtained using the Kaplan–Meier method. Survival rates between patients treated with tocilizumab within two days after hospital admission and later, and between omicron and delta cases, were compared using the log-rank test. Then, survival rates between patients treated with tocilizumab + SoC (studied population) and SoC alone (control group), and between patients treated with tocilizumab within two days after hospital admission and later (studied population divided in two groups), and those treated with SoC alone (control group) were also compared using the log-rank test.

All tests were two-sided, and a P-value less than 0.05 was considered statistically significant. Analyses were performed using the R software version 4.2.0 (The R Foundation for Statistical Computing, Vienna, Austria).

## 5. Conclusions

In a cohort of severe COVID-19 patients infected with delta and omicron VOC and treated with tocilizumab in a French teaching hospital, we reported a 22% mortality. Mortality rates were similar in the omicron and delta cases, and in COVID-19 patients treated with tocilizumab plus SoC and SoC alone. However, our results suggested that the early administration of tocilizumab had a protective effect on mortality in severe COVID-19 patients regardless of the VOC involved. In contrast, a higher age, higher Charlson index and higher WHO-CPS were significantly associated with mortality. Further multicenter studies are awaited to confirm our results.

## Figures and Tables

**Figure 1 antibiotics-12-00088-f001:**
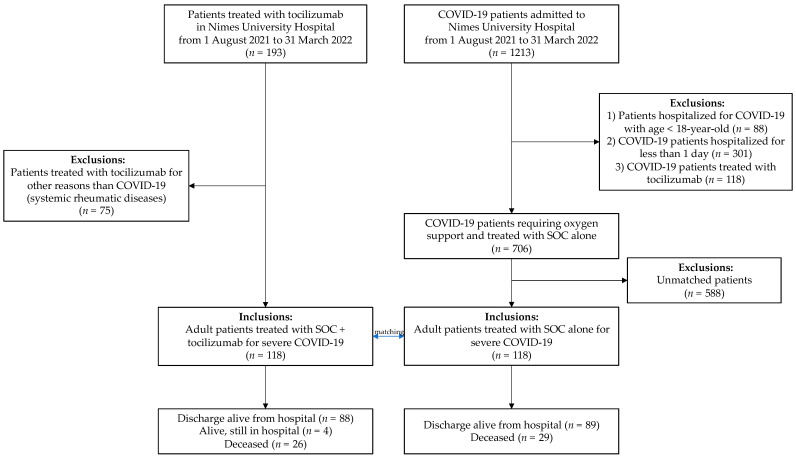
Flow chart of the study population, depicting the selection of patients treated with tocilizumab for severe COVID-19 and the selection of patients with COVID-19 not treated with tocilizumab (matched control group on age, sex, oxygen needs, Charlson index and variant of concern).

**Figure 2 antibiotics-12-00088-f002:**
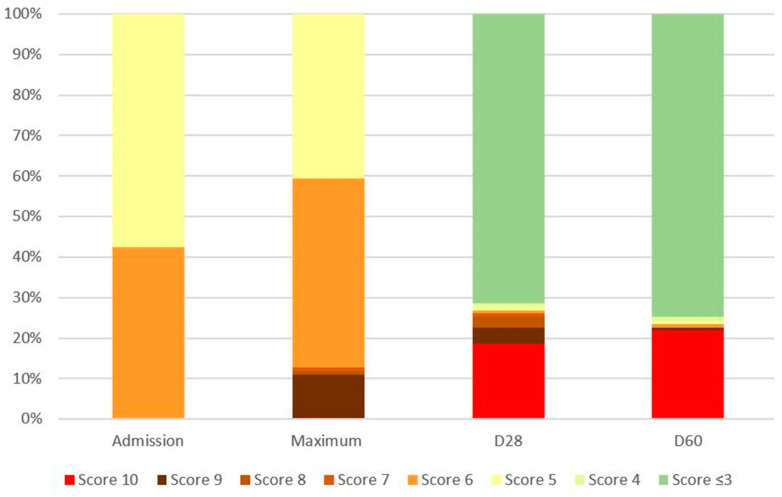
World Health Organization Clinical Progression Scale (WHO-CPS) evolution during the 60-day follow-up. (D28 and D60: assessment of WHO-CPS 28 and 60 days after admission).

**Figure 3 antibiotics-12-00088-f003:**
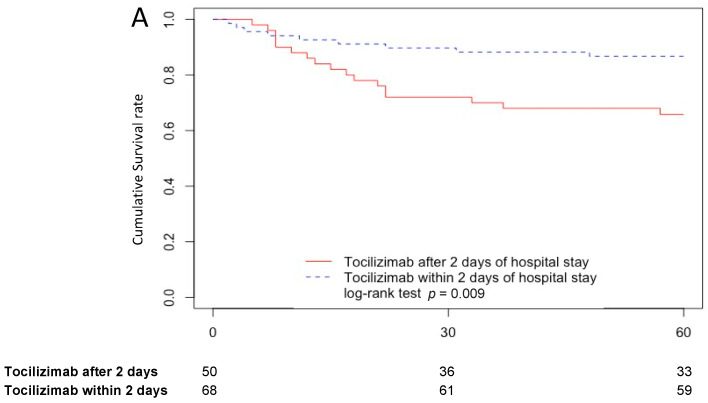
Kaplan–Meier Survival Curves at day 60 for patients treated with tocilizumab within 2 days of hospital admission and patients treated with tocilizumab after 2 days (**A**), in omicron and delta cases treated with tocilizumab (**B**), in patients treated with standard of care (SoC) alone and with SoC plus tocilizumab (**C**), in patient treated with SoC alone and patients treated with tocilizumab plus SoC within 2 days of hospital admission and patients treated with tocilizumab plus SoC after 2 days (**D**).

**Table 1 antibiotics-12-00088-t001:** Patients’ characteristic and outcomes.

Characteristics	Total (N = 118)	Survivor (N = 92)	Non-Survivor (N = 26)	*p*-Value
Age, years	69 (56–77)	64 (54–72)	78 (71–84)	<0.001 *
Male	79 (67%)	61 (66%)	18 (69%)	0.78
BMI, kg/m^2 a^	28 (25–31)	28 (25–31)	27 (24–31)	0.20
Main comorbidities				
Diabetes	36 (31%)	24 (26%)	12 (46%)	0.05
Myocardial ischemia	18 (15%)	9 (10%)	9 (35%)	0.003 *
Chronic heart failure	5 (4%)	2 (2%)	3 (12%)	0.06
Chronic lung disease	18 (15%)	12 (13%)	6 (23%)	0.22
Chronic kidney disease	7 (6%)	4 (4%)	3 (12%)	0.19
Dementia	5 (4%)	1 (1%)	4 (15%)	0.01 *
Cancer	19 (16%)	11 (12%)	8 (31%)	0.03 *
Hemopathy ^b^	8 (7%)	4 (4%)	4 (15%)	0.06
Charlson index	1 (0–2)	1 (0–2)	2.5 (1–5)	<0.001 *
Chest CT-Scan ^c^				
Mild	13 (11%)	8 (9%)	5 (19%)	0.14
Moderate	72 (61%)	59 (64%)	13 (50%)	0.2
Severe	24 (20%)	18 (20%)	6 (23%)	0.7
Critical	8 (7%)	6 (7%)	2 (8%)	0.83
Variant of concern				
Delta	101 (86%)	78 (85%)	23 (88%)	0.64
Omicron	17 (14%)	14 (15%)	3 (12%)	0.64
Oxygen requirement				
Conventional oxygen therapy	43 (36%)	41 (45%)	2 (8%)	0.003 *
High-flow nasal canula	57 (48%)	39 (42%)	18 (69%)	0.02 *
Mechanical ventilation	18 (15%)	12 (13%)	6 (23%)	0.22
ECMO ^d^	6 (5%)	5 (5%)	1 (4%)	0.75
WHO-CPS ^e^ at admission	5 (5–6)	5 (5–6)	5 (5–6)	0.66
WHO-CPS at tocilizumab administration	6 (5–6)	5 (5–6)	6 (5–6)	0.047 *
Tocilizumab treatment				
Administration timing, days	2 (2–4)	2 (2–3)	3.5 (2–5)	0.002 *
Administration within 2 days	68 (58%)	59 (64%)	9 (35%)	0.009 *
Inflammation biomarker				
CRP pre-tocilizumab, mg/L	124 (89–165)	128 (91–166)	103 (77–162)	0.45
CRP post-tocilizumab, mg/L	47 (23–89)	47 (23–81)	54 (24–95)	0.25
Outcomes				
Death at day 60	26 (22%)	-	-	
ICU ^f^ admission	56 (47%)	39 (42%)	17 (65%)	0.04 *
Limitation of life support	20 (17%)		20 (77%)	
CAPA ^g^	3 (3%)	1 (<1%)	2 (8%)	0.1

Results are expressed as median and interquartile range (IQR) or as number of patients and percentage (%) as appropriate. ^a^ BMI: body mass index, ^b^ Hemopathy: lymphoma, leukemia or myeloma. ^c^ Chest CT-Scan were classified according to percentage of pulmonary involvement: Mild (10–25%); Moderate (25–50%); severe (50–75%) and critical (75–100%), ^d^ ECMO: extracorporeal membrane oxygenation, ^e^ WHO-CPS: World Health Organization Clinical Progression Scale, ^f^ ICU: intensive care unit, ^g^ CAPA: COVID-19 associated pulmonary aspergillosis. * Statistically significant.

**Table 2 antibiotics-12-00088-t002:** Factors associated with mortality in patients treated with tocilizumab.

	Univariable Analysis			Multivariable Analysis		
Characteristics	HR	95% CI	*p*-Value	HR	95% CI	*p*-Value
Male	1.2	0.5–2.7	0.73			
Age	1.1	1–1.1	<0.001 *	1.06	1.02–1.11	0.002 *
BMI	0.95	0.88–1	0.2 *	0.99	0.91–1.08	0.83
Charlson index	1.5	1.3–1.7	<0.001 *	1.33	1.11–1.6	0.002 *
Chest CT-Scan ^a^						
Mild	2.1	0.78–5.5	0.14 *	0.99	0.31–3.12	0.98
Moderate	0.61	0.28–1.3	0.21			
Severe	1.2	0.47–2.9	0.74			
Critical	1.3	0.3–5.3	0.76			
Variant of concern						
Delta	1.3	0.4–4.4	0.65			
Omicron	0.76	0.23–2.5	0.65			
WHO-CPS ^b^						
At admission	1.2	0.54–2.5	0.69			
At tocilizumab administration	2.3	0.98–5.2	0.055 *	2.58	1.07–6.22	0.03 *
Inflammation biomarkers						
CRP pre-tocilizumab	1	0.99–1	0.49			
CRP post-tocilizumab	1	1–1	0.16 *	1	1–1.01	0.27
Tocilizumab treatment						
Administration within 2 days	0.35	0.16–0.73	0.012 *	0.37	0.14–0.97	0.04 *
Dose	1	0.99–1	0.16 *	1	0.99–1	0.57

^a^ Chest CT-Scans were classified according to percentage of pulmonary involvement: Mild (10–25%); Moderate (25–50%); severe (50–75%) and critical (75–100%), ^b^ WHO-CPS: World Health Organization Clinical Progression Scale. * Statistically significant.

## Data Availability

The authors consent to share the collected data with others. The raw data supporting the conclusions of this article will be made available by the authors, without undue reservation. Data will be available immediately after the main publication and indefinitely.

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
