# Peer review of "Effect of Tocilizumab on Mortality in Patients with SARS-CoV-2 Pneumonia Caused by Delta or Omicron Variants: A Propensity-Matched Analysis in Nimes University Hospital, France"

_antibiotics, 2023, doi:10.3390/antibiotics12010088_

Round 1
Reviewer 1 Report
Congratulations for the work done
I analysed the paper and, from my review, I make the following observations:
In general, it would be useful to improve English and correct some typos and it's useful to insert the full words before entering acronyms.
Title: add the name of the location of the study
Keywords: add mortality rates
Methods: improve the explanation about the patients and about the use (normal and off topics) of Tocilizumab.
Statistical analysis: add and explain better the data about survival curves.
Conclusion: Improve and increase the conclusions.
References: It would be useful to insert some article on covid management, such as: “An Emerging Innovative UV Disinfection Technology (Part II): Virucide Activity on SARS-CoV-2”. Gabriele Messina, Alessandro Della Camera, Pietro Ferraro, Davide Amodeo, Alessio Corazza, Nicola Nante, Gabriele Cevenini. Int J Environ Res Public Health. 2021 Apr 7;18(8):3873. doi: 10.3390/ijerph18083873
Author Response
Congratulations for the work done
I analysed the paper and, from my review, I make the following observations:
First, we thank the reviewer for their interest to this work and for their observations.
In general, it would be useful to improve English and correct some typos and it's useful to insert the full words before entering acronyms.
The manuscript had benefit from a language check by a colleague who is a native English speaker.
Title: add the name of the location of the study
The title has been amended with the location of the study (Nimes University Hospital)
Keywords: add mortality rates
We added “mortality rates” in the keywords as suggested by the reviewer
Methods: improve the explanation about the patients and about the use (normal and off topics) of Tocilizumab.
As requested by the reviewer, we improved the description of patient selection and tocilizumab use in the method section. Please see method section “tocilizumab treatment” and “patients” paragraphs.
Statistical analysis: add and explain better the data about survival curves.
Done. Please see the paragraph “statistical analysis”
Conclusion: Improve and increase the conclusions.
Done. Please see Conclusions section.
References: It would be useful to insert some article on covid management, such as: “An Emerging Innovative UV Disinfection Technology (Part II): Virucide Activity on SARS-CoV-2”. Gabriele Messina, Alessandro Della Camera, Pietro Ferraro, Davide Amodeo, Alessio Corazza, Nicola Nante, Gabriele Cevenini. Int J Environ Res Public Health. 2021 Apr 7;18(8):3873. doi: 10.3390/ijerph18083873
We thank the reviewer for this suggestion. However, we did not understand in which part of the paper we can add such a citation.
Maybe the reference is not what the reviewer wanted to suggested? We believe, perhaps wrongly, that the cited article is too general to be part of an article focusing on one of the immunomodulatory treatments of COVID-19. Please note that guidelines for management of COVID-19 patients have been cited.
Reviewer 2 Report
The manuscript is well-written and scientifically sound. I would suggest that the authors consider the following points as they revise their manuscript and continue their work in this important research area.
Please add specific keywords, do not use the same word keywords and title.
The objectives of this study are absent. After the introduction of the associated research, one paragraph for the objectives statement is necessary.
The introduction and Discussion need minor revision. The author could check a recently published article.
Author Response
The manuscript is well-written and scientifically sound. I would suggest that the authors consider the following points as they revise their manuscript and continue their work in this important research area.
The authors thank the reviewer for their interest in our work and for their remarks.
Please add specific keywords, do not use the same word keywords and title.
As suggested by the reviewer we used keywords different from the title. Pease see keywords p1.
The objectives of this study are absent. After the introduction of the associated research, one paragraph for the objectives statement is necessary.
We written a specific paragraph for the objectives of the study. Please see the end of introduction p2.
The introduction and Discussion need minor revision. The author could check a recently published article.
The introduction and discussion sections have been updated as recommended by the reviewer.
Reviewer 3 Report
Authors tried to find the significance of Tocilizumab on mortality in patients with SARS-CoV-2 2 pneumonia due to delta or omicron variants:. I like the work and it is of highest importance. The paper is well written, but I have some suggestions that may help to improve the paper's soundness and novelty.
- The introduction section should be improved by including a brief description of omicron variants and Tocilizumab mechanism of action against omicron.
- The study needs to mention the name and location of the ethical/review board of the hospital and patient consent.
- Justify the sample size and how is it significant?
- What are the study's selection criteria?Is there any significance or conclusion about why the mortality rate was so high after 60 days, when it was 22%?
- The parameters for the plus standard of 54-hour care (SoC) were mentioned.
- Is there a significant conclusion about why the mortality rate was so high in the case of a 60-day period with a 22% mortality rate?Explain why you're describing 60-day data.
- Line number 47Take out the full stop...the approval (December 2021 in Europe)…
- Is it 312% in table 1 under line omicron? This needs to be confirmed.
- Is there any conclusive evidence that non-servivors have a lower (23%) risk of severe lung infection than diabetics (46%).[If data is available, please include a reference.]
- Improve the conclusion
Author Response
Reviewer 3
Authors tried to find the significance of Tocilizumab on mortality in patients with SARS-CoV-2 2 pneumonia due to delta or omicron variants:. I like the work and it is of highest importance. The paper is well written, but I have some suggestions that may help to improve the paper's soundness and novelty.
We thank the reviewer for their thorough comments and interest in our work.
- The introduction section should be improved by including a brief description of omicron variants and Tocilizumab mechanism of action against omicron.
The introduction has been updated as recommended by the reviewer.
- The study needs to mention the name and location of the ethical/review board of the hospital and patient consent.
As requested by the reviewer, we added the name and location of the ethical/review board of the hospital and patient consent. Please see lines 296-302, p9
- Justify the sample size and how is it significant?
As the reviewer correctly pointed out, the sample size is actually too small to detect a difference in mortality between patients treated with tocilizumab + SoC and those treated with SoC alone. Indeed, we should have included 3902 patients in the study, 1951 in each group, with an alpha risk=0.05 and a power at 90%, to detect a 4% reduction in mortality (corresponding to the reduction in mortality rate reported in the RECOVERY trial). Our study is exploratory and our results suggest that tocilizumab is feasible in delta and omicron cases, and the interest of early administration. We acknowledge this limit. Please see limit paragraph in the discussion section, p9
- What are the study's selection criteria? Is there any significance or conclusion about why the mortality rate was so high after 60 days, when it was 22%?
As mentioned in the method section, paragraph “tocilizumab”, the use of tocilizumab was let at physicians’ discretion, and based on local guidelines. The selection criteria included inthese guidelines were as follow: 1) CRP ≥75 mg/L and an increase of oxygen needs ≥ 2L/min within 48 hours, or 2) lack of improvement after 48 hours of SoC, or 3) ICU admission ≤ 24 hours. Taking into account these criteria, we probably included more severe patients than in the general population infected by the COVID-19. However, and despite our population is older than those included in clinical trials, mortality rates were similar to those reported in clinical trials. Please see limit paragraph in the discussion section, p9.
- The parameters for the plus standard of 54-hour care (SoC) were mentioned.
We apologize. We did not understand the question / remark of the reviewer.
The SoC are mentioned in the method section. Please see p10.
- Is there a significant conclusion about why the mortality rate was so high in the case of a 60-day period with a 22% mortality rate?Explain why you're describing 60-day data.
As previously discussed, our population was older than those of many studies, and patients were more severe. However, the mortality rates seemed in line with those reported in the literature (RECOVERY trial ≥30% at day-28, REMAPCAP 22-28% at hospital discharge and >30% at day-180), taking into account the case mix of delta and omicron VOCs.
The respiratory and vital status were assessed at 28 and 60 days after hospital admission as recommended by the WHO working group on the clinical characterization and management of COVID-19 infection (WHO Working Group on the Clinical Characterisation and Management of COVID-19 infection, Lancet Infect Dis 2020)
- Line number 47Take out the full stop...the approval (December 2021 in Europe)…
Done
- Is it 312% in table 1 under line omicron? This needs to be confirmed.
We thank the reviewer for pointing out this mistake. One parenthesis was missing: 3 (12%). We amended the Table 1 accordingly. Please see Table 1.
- Is there any conclusive evidence that non-servivors have a lower (23%) risk of severe lung infection than diabetics (46%).[If data is available, please include a reference.]
We agree with the reviewer that the excess mortality in diabetic patient from COVID-19 is a important concern. We amended the discussion accordingly (please see the second paragraph of the discussion) and cited the following reference: Excess diabetes mellitus-related deaths during the COVID-19 pandemic in the United States, Fan Lv et al. EClinicalMedicine, 2022.
- Improve the conclusion
As suggested by the reviewer we improved the content of the conclusion. Please see conclusions section.
Reviewer 4 Report
I appreciate the opportunity to review the manuscript entitled "Effect of Tocilizumab on mortality in patients with SARS-CoV- 2 pneumonia due to delta or omicron variants: a single-centre propensity-matched analysis". In the current prospective cohort study by Paul Laffont-Lozes et al. have evaluated the protective effect on COVID-19 induced mortality.
However, the author may take note of the major and minor remarks listed below to improve the manuscript:
Major comments:
- The manuscript is poorly written and requires extensive corrections for grammatical errors, consistency, and wordiness. For e.g.,
· Line no: 27: ‘full stop’ given in continuous sentence
· Line no: 29: ‘were’ should be used instead of ‘was’
· Line no: 43: to improve readability instead of ‘due to’ the word ‘caused by’ should be written
· Line no: 47: multiple ‘full stops’ added
· Line no: 76: the phrase used in sentence “Among the study population the median hospital length of stay was 10 days (IQR, 7- 15)” is wrong.
· The full form of ‘CAPA’ should be mentioned in earlier section.
2. Figure legends should be self-explanatory for e.g.,
· Instead of mentioning only p values in table 1 legend author should describe the groups in which they found significant difference.
· ‘Figure 1. Flow chart’ is general term which does not provide any essential information.
3. In outcome section authors have mentioned the observed CRP values. However, the proportion of patients with increased CRP before and after tocilizumab is not mentioned on result section and not discussed in discussion.
4. In the second paragraph of the discussion section authors should compare and discuss the effect of the tocilizumab in the patients infected with alpha, beta, gamma, delta, and omicron variants.
5. Did those 6 patients who had received second dose of tocilizumab cited studies (2 and 26) survived? Authors should discuss the observed outcomes.
Miner comments:
6. Mention full form of WHO-CPS
7. Line no: 214: what is uncontrolled
Author Response
I appreciate the opportunity to review the manuscript entitled "Effect of Tocilizumab on mortality in patients with SARS-CoV- 2 pneumonia due to delta or omicron variants: a single-centre propensity-matched analysis". In the current prospective cohort study by Paul Laffont-Lozes et al. have evaluated the protective effect on COVID-19 induced mortality.
However, the author may take note of the major and minor remarks listed below to improve the manuscript:
We thank the reviewer for their helpful remarks and comments, and interest in our work.
Major comments:
- The manuscript is poorly written and requires extensive corrections for grammatical errors, consistency, and wordiness.
The manuscript had benefit from a language check by a colleague who is a native English speaker.
For e.g.,
- Line no: 27: ‘full stop’ given in continuous sentence
The full stop has been removed.
- Line no: 29: ‘were’ should be used instead of ‘was’
Done
- Line no: 43: to improve readability instead of ‘due to’ the word ‘caused by’ should be written
Done
- Line no: 47: multiple ‘full stops’ added
Multiple full stops have been removed.
- Line no: 76: the phrase used in sentence “Among the study population the median hospital length of stay was 10 days (IQR, 7- 15)” is wrong.
We reworded the sentence in “the median hospital length of stay was 10 days (IQR, 7- 15)”
- The full form of ‘CAPA’ should be mentioned in earlier section.
Done. Please see line 173, p4
- Figure legends should be self-explanatory for e.g.,
- Instead of mentioning only p values in table 1 legend author should describe the groups in which they found significant difference.
Done
- ‘Figure 1. Flow chart’ is general term which does not provide any essential information.
As suggested by the reviewer the Figure legends have been improved.
- In outcome section authors have mentioned the observed CRP values. However, the proportion of patients with increased CRP before and after tocilizumab is not mentioned on result section and not discussed in discussion.
We thank the reviewer for this remark. Most of the patients had a decrease in CRP values after tocilizumab except for 9 of them. We amended the Results section accordingly. We also discuss the impact of CRP decrease (or not) and its interest in monitoring the effect of tocilizumab in the discussion (please see 5th paragraph of the discussion section, p8)
- In the second paragraph of the discussion section authors should compare and discuss the effect of the tocilizumab in the patients infected with alpha, beta, gamma, delta, and omicron variants.
Done
- Did those 6 patients who had received second dose of tocilizumab cited studies (2 and 26) survived? Authors should discuss the observed outcomes.
Once again, we thank the reviewer for this remark. All the 6 patients who received a second dose of tocilizumab survived. Please see 5th paragraph of the discussion section, p8
Miner comments:
- Mention full form of WHO-CPS
As suggested by the reviewer we mentioned the full form of WHO-CPS. Please see Table 1, Table 2 and line 130 p3.
- Line no: 214: what is uncontrolled
We reworded the sentence in “untreated” infections. See line 1095.
Round 2
Reviewer 4 Report
Dear Authors,
Thank you for incorporating the suggested changes.
Namdev